# Integrating Machine Learning-Based Approaches into the Design of ASO Therapies

**DOI:** 10.3390/genes16020185

**Published:** 2025-02-02

**Authors:** Jamie Leckie, Toshifumi Yokota

**Affiliations:** 1Department of Medical Genetics, Faculty of Medicine and Dentistry, University of Alberta, Edmonton, AB T6G 2H7, Canada; jleckie@ualberta.ca; 2The Friends of Garrett Cumming Research & Muscular Dystrophy Canada HM Toupin Neurological Sciences Research, Edmonton, AB T6G 2H7, Canada

**Keywords:** antisense oligonucleotides (ASOs), machine learning, design, eSkip-Finder, ASOptimizer, rare disease

## Abstract

Rare diseases impose a significant burden on affected individuals, caregivers, and healthcare systems worldwide. Developing effective therapeutics for these small patient populations presents substantial challenges. Antisense oligonucleotides (ASOs) have emerged as a promising therapeutic approach that targets the underlying genetic cause of disease at the RNA level. Several ASOs have gained FDA approval for the treatment of genetic conditions, including use in personalized N-of-1 trials. However, despite their potential, ASOs often exhibit limited clinical efficacy, and optimizing their design is a complex process influenced by numerous factors. Machine learning-based platforms, including eSkip-Finder and ASOptimizer, have been developed to address these challenges by predicting optimal ASO sequences and chemical modifications to enhance efficacy. eSkip-Finder focuses on exon-skipping applications, while ASOptimizer aims to optimize ASOs for RNA degradation. Preliminary in vitro results have demonstrated the promising predictive power of these platforms. However, limitations remain, including their generalizability to alternative targets and gaps in their consideration of all factors influencing ASO efficacy and safety. Continued advancements in machine learning models, alongside efforts to incorporate additional features affecting ASO efficacy and safety, hold significant promise for the field. These platforms have the potential to streamline ASO development, reduce associated costs, and improve clinical outcomes, positioning machine learning as a key tool in the future of ASO therapeutics.

## 1. Introduction

Rare diseases, commonly referred to as “orphan diseases”, are defined by a prevalence of fewer than 1 in 2000 individuals [1]. Although each disease affects only a small number of people, there are approximately 8000 identified rare diseases, collectively impacting an estimated 3.5–5.9% of the global population [2,3]. Around 80% of these diseases are attributed to genetic causes, many of which manifest in early stages of life and can be associated with reduced survival [4,5,6]. In the United States alone, the economic burden of rare diseases, encompassing both direct medical expenses and indirect costs, is estimated to approach USD 1 trillion annually [7,8]. The limited patient populations for rare diseases pose significant challenges in developing treatments and conducting meaningful clinical trials [9]. Pharmaceutical companies are often reluctant to invest in these treatments due to the low expected revenues from such a small patient group. Although legislative initiatives and government funding have enhanced efforts toward rare disease therapeutics [9,10], additional challenges remain. These include the use of suboptimal endpoints, lack of appropriate controls, and difficulties in recruiting patients for clinical trials [11,12]. Consequently, only 5% of patients with rare diseases currently have access to potentially disease-modifying treatments [13]. There is a critical need to improve processes for researching and developing therapeutics for rare diseases, accelerating their time to market while reducing associated costs.

Antisense oligonucleotides (ASOs) have emerged as promising therapeutic strategies for rare diseases due to their versatility [14]. By modulating the splicing or degradation of target RNA with high specificity, ASOs hold the potential to address a broad range of rare diseases [15]. Several ASOs have already received FDA approval for treating rare genetic conditions [16,17,18,19]. Furthermore, collaborative efforts with the FDA have enabled multiple patients to access personalized N-of-1 ASOs, which has subsequently led to the establishment of guidelines for N-of-1 ASO development [20]. Despite these advances, there is still significant room to improve the clinical efficacy of ASOs. Designing optimal ASOs is intrinsically complex, requiring careful consideration of a vast array of potential sequences and numerous factors that impact their safety and effectiveness [21]. Historically, identifying the most effective sequence and chemistry relied on labor-intensive and time-consuming experimental trials [22,23,24,25]. However, the importance of optimizing ASO design has become increasingly evident, as ASOs with optimized target sequences have demonstrated greater efficacy than many currently FDA-approved counterparts [25,26,27].

To leverage the extensive experimental data available on ASO efficacy, encompassing parameters such as target sequence and chemical modifications, machine learning-based platforms have been developed [21,28]. These platforms hold significant potential to predict the efficacy of novel ASOs, paving the way for the identification of more effective therapies [29]. A previous review primarily focused on providing an overview of a single machine learning-based platform [29]. In contrast, this review provides a comprehensive overview of ASOs, their application in N-of-1 trials, the challenges associated with optimizing ASO design, and current machine learning-based platforms that have been developed in an attempt to improve ASO design. Apart from a brief introduction covering the history of the first FDA-approved ASO, the focus is specifically on short ASOs targeting genetic diseases. Alternative RNA-based therapeutics, such as siRNAs and mRNA vaccines, operate on distinct design principles and mechanisms of action and are, therefore, beyond the scope of this review.

## 2. Antisense Oligonucleotides (ASOs)

Initially discovered in 1978 [30], ASOs are single-stranded, RNA- or DNA-like molecules that are capable of regulating gene expression through a range of mechanisms [31]. They are typically composed of between 12 and 25 nucleotides that are complimentary to a specific region of the target mRNA and/or pre-mRNA. Through Watson–Crick pairing, ASOs bind to their targets, where, with high specificity, they can regulate RNA processing and translation [15].

### 2.1. Chemical Modifications

Although these versatile molecules are highly specific, they face significant challenges in penetrating biological membranes to reach target cells in vivo when they are unmodified [32]. In addition, the presence of the phosphodiester bond on unmodified ASOs causes them to be relatively unstable due to degradation by nucleases [33]. To improve their stability, safety, and overall efficacy, numerous chemical modifications have been identified that can be applied to some or all nucleotides, depending on the desired mechanism of the ASO. The first chemical modification utilized to improve ASO stability, known as phosphorothioates (PS) ASOs, was made by replacing the non-bridging oxygen of the phosphodiester bond with a sulfur group [34]. PS ASOs have a significantly longer half-life than unmodified ASOs due to their ability to evade degradation by nucleases [35,36]. However, the PS modifications reduce an ASO’s binding affinity for its target, which can be improved by combining PS modifications with modifications to the sugar moiety [37]. ASOs commonly used possessing chemical modifications to their sugar component include locked nucleic acids (LNAs), 2′-O-methyl (2′-OME), 2′-O-methoxyethyl (2′-MOE), and 2′fluoro ASOs. LNA modifications are composed of a methylene link that creates a bridge between the 2′ oxygen and 4′ carbon of the sugar moiety, which locks the nucleotide in a conformation that substantially increases the ASO’s affinity for its target RNA [38]. Alternatively, 2′-OME, 2′-MOE, and 2′fluoro ASOs are created through modifications to the 2′ position of the ribose ring [39]. These 2′ substituents promote the molecule’s stability by protecting it from nuclease degradation, as well as improving its target affinity [40,41]. 2′-OME and 2′-MOE modifications are also associated with improved safety due to limited immune activation [15]. Additional chemical modifications, including phosphorodiamidate morpholino oligonucleotides (PMOs) and peptide nucleic acids (PNAs), result in a more substantial change to the overall nucleotide chemistry. Specifically, PMOs are charge-neutral ASOs that utilize a six-membered morpholine ring in place of the natural five-membered ribose ring, linked through phosphorodiamidate bonds [42]. PNAs are composed of an N-2-aminoethyl backbone [43]. Both PMOs and PNAs are resistant to nuclease degradation and have improved target binding affinity [44,45]. The neutral charge uniquely associated with these ASOs improves their safety by reducing immune stimulation and allows them to be conjugated to cell-penetrating-peptides for the potential to improve delivery [37,46].

### 2.2. Mechanisms of Action

ASOs are highly versatile therapeutic tools capable of employing various mechanisms to mitigate the effects of a wide range of pathogenic mutations. These mechanisms can be broadly classified into two primary categories: RNA degradation (RNase H-dependent) and steric blockage (RNase H-independent) [15]. Depending on the desired mechanism of action, ASOs can be chemically modified and designed to target specific RNA regions to achieve their intended therapeutic effects [47,48].

RNase H-dependent RNA degradation is particularly effective for addressing diseases caused by gain-of-function mutations, where reducing or eliminating the production of the mutated protein can improve the disease phenotype [49]. RNase H is an endogenous enzyme that plays an important role in cleaving RNA when it becomes hybridized to DNA [50]. ASOs, due to their DNA-like qualities, exploit this natural mechanism to target and degrade disease-causing mRNA (Figure 1). To enable RNase H-mediated degradation, ASOs must retain some unmodified regions, as full chemical modification prevents recognition by RNase H [51,52]. However, PS modifications can be applied to all nucleotides in an ASO without compromising RNase H recognition [53]. To balance the advantages of additional chemical modifications with the need for RNase H activity, a specialized ASO design known as a gapmer is used. In this configuration, the central region of the ASO remains unmodified to allow RNase H recognition, while the flanking regions incorporate alternative chemical modifications for enhanced stability and efficacy [54,55].

In cases where a disease arises from a loss-of-function mutation that disrupts the production of functional proteins, ASOs can regulate pre-mRNA splicing to compensate for these pathogenic mutations [56]. These ASOs are designed to bind to specific regions of pre-mRNA typically recognized by splicing factors. By sterically blocking the binding of splicing machinery, ASOs can modulate the inclusion or exclusion of specific exons to achieve the desired therapeutic effect [57]. ASO-mediated exon-skipping has received significant interest due to its ability to exclude cryptic exons, exons containing pathogenic mutations, exons adjacent to large out-of-frame deletions, or poison exons from the final mRNA transcript, offering a powerful approach to treating genetic diseases [58,59,60]. Additionally, ASOs can promote exon inclusion, although this strategy presents greater technical challenges, as identifying effective target regions to promote exon inclusion is more complex [61]. To avoid RNase H-mediated degradation of RNA, ASOs designed for steric blockage are typically fully chemically modified to not be detected by RNase H [62,63]. These fully modified ASOs have also proven effective in inhibiting translation by sterically blocking the interaction of translation machinery with the bound RNA, further broadening their therapeutic potential [64,65].

### 2.3. FDA-Approved ASOs

There are currently many ASOs that have received FDA approval, utilizing different chemical modifications and mechanisms of action to treat genetic diseases (Table 1). Fomiversen provided the first clinical proof of concept of the effectiveness of ASOs in patients and was approved for the treatment of cytomegalovirus (CMV) retinitis, particularly for patients with acquired immune deficiency syndrome (AIDS) [16]. It is composed of 21 nucleotides with PS backbone modifications, targeting the major immediate-early (*MIE*) gene of the virus to prevent the translation of critical proteins for its replication [66,67]. Through intravitreal injections, Fomiversen was found to effectively prevent the progression of CMV retinitis in affected patients, with a tolerable safety profile [68,69]. Despite its promising results in early clinical studies, the approval of highly active antiretroviral therapy (HAART) led to its FDA approval being withdrawn in 2001, as HAART was effective in curing or preventing CMVR in patients with AIDS [70,71].

In recent years, additional ASOs utilizing RNase H-dependent RNA degradation have received FDA approval for the treatment of disease. Mipomersen, approved in January 2013, is designed for homozygous familial hypercholesterolemia (HoFH), a genetic condition characterized by an impaired ability to clear low-density lipoproteins (LDL), placing patients at high risk for heart disease [17]. Mipomersen is a PS gapmer with 2′-MOE-modified ends, targeting *ApoB-100* mRNA for RNase H-mediated degradation [72]. By reducing *ApoB* translation, Mipomersen is effective in reducing very-low-density lipoprotein and subsequently lowering the levels of circulating LDLs [73]. Similarly, Inotersen, also a PS gapmer with 2′-MOE-modified ends, targets transthyretin (*TTR*) mRNA to prevent the production of dysfunctional TTR proteins, which accumulate as deposits causing sensorimotor and autonomic neuropathy in hereditary transthyretin amyloidosis (hATTR) [74,75]. Inotersen received FDA approval in October 2018 [74]. Most recently, Tofersen, employing the same chemical modifications as Mipomersen and Inotersen, was approved in April 2023 for amyotrophic lateral sclerosis (ALS) associated with *SOD1* gene mutations [76].

Alternatively, several ASOs designed to regulate RNA splicing have been FDA approved for treating rare diseases. Among these, four ASOs with PMO chemical modifications, designed to enhance stability and safety, have been approved for Duchenne muscular dystrophy (DMD) [18]. DMD typically arises from mutations in the *DMD* gene that disrupt the reading frame, preventing the production of functional dystrophin [77,78]. Patients with in-frame mutations, including large deletions spanning multiple exons, often present with the milder Becker muscular dystrophy phenotype [79]. ASOs can restore the reading frame by promoting the exclusion of exons flanking the disrupted region, allowing for the production of truncated but partially functional dystrophin [80]. Since these DMD-causing frame-shift mutations occur throughout the *DMD* gene [81,82], each exon-skipping approach targets specific subsets of DMD patients [83]. Eteplirsen, targeting exon 51, became the first ASO approved for DMD in September 2016 [19]. This was followed by the approvals of Golodirsen and Viltolarsen, both targeting exon 53, and Casimersen, targeting exon 45 [84].

Splice-switching ASOs have also been approved for spinal muscular atrophy (SMA), a severe rare disease caused by pathogenic mutations in the *SMN1* gene [85]. Unlike ASOs for DMD, which target the mutated gene, these ASOs modulate the splicing of *SMN2*, a pseudogene that differs from *SMN1* by a single C-to-T transition in exon 7, leading to its exclusion in most *SMN2* transcripts [86]. Nusinersen, approved in December 2016, promotes the inclusion of exon 7 in *SMN2* transcripts, effectively compensating for the loss of *SMN1* function [42]. As this therapeutic targets *SMN2* pre-mRNA, its applicability is independent of the specific mutations in the *SMN1* gene.

### 2.4. ASOs as N-of-1 Therapies

Recent advances in N-of-1 therapeutics, designed to target disease-causing mutations unique to a single patient or very small groups, have provided hope for individuals with ultra-rare genetic conditions. ASOs have emerged as a promising approach for such cases due to their high specificity, ability to address the root cause of disease, and demonstrated safety profile [15]. With their growing popularity and recent FDA approvals, the processes for preclinical ASO testing and evaluating relevant potential adverse effects are now well established [87]. As the focus on personalize medicine grows and regulations evolve to enable single-patient clinical trials, ASOs have started to become accessible to patients through N-of-1 trials (Table 2).

Milasen represents a landmark in N-of-1 therapeutics as the first ASO designed to treat a single patient [88]. Mila, a 6-year-old girl at the time of the study in 2017, was experiencing a rapid health decline, including sudden blindness, ataxia, seizures, and developmental delay. Genetic testing revealed that she carried two distinct mutations in each allele of the *CLN7* gene, associated with Batten disease [89]. Batten disease is associated with the accumulation of lipofuscin in the nervous system, leading to severe neurological symptoms and premature death [90]. One mutation Mila possessed had been previously reported. The other mutation was a unique SINE-VNTR-*Alu* (SVA) insertion in intron 6, which caused the inclusion of a cryptic intronic sequence in the final mRNA product [91], resulting in a dysfunctional protein. Milasen, a 22-nucleotide 2′-MOE-modified ASO, was designed to target the intron 6 cryptic splice-acceptor site. Preclinical in vitro and in vivo studies demonstrated that Milasen improved the ratio of wild-type *CLN7* mRNA [88]. With Mila’s health beginning to deteriorate, the N-of-1 trial was approved by the FDA, following a dosing regimen like Nusinersen due to similarities in chemistry, target tissue, and mechanism of action. While Milasen significantly reduced the frequency of Mila’s seizures without adverse effects, she ultimately succumbed to Batten disease three years after treatment began. Despite this outcome, the rapid development of Milasen, completed within a year of identifying Mila’s unique variant, set a powerful precedent for the use of ASOs in personalized medicine for rare diseases [88,92].

Following the success of Milasen, the same team developed Atipeksen, another N-of-1 ASO, for a 1-year-old girl named Ipek diagnosed with severe ataxia–telangiectasia (A-T) [93]. A-T, also known as Louis Bar syndrome, is a rare autosomal recessive disease caused by mutations in the *ATM* gene [94], leading to progressive neurodegeneration, immune dysfunction, and significantly shortened life expectancy [95]. Ipek was found to have compound heterozygous *ATM* mutations, one of which created a splice donor site in exon 53 that prevented the entire exon 53 from being included in the final transcript, resulting in a frameshift [93]. To restore normal splicing, 32 novel 2′-MOE-modified ASOs were evaluated, targeting either the aberrant splice donor site or an adjacent splice silencer site. After further functional testing, a single ASO, named Atipeksen, was identified as the most effective candidate. Following animal safety studies, Atipeksen was approved for the clinical trial in Ipek and was administered intracerebroventricularly every three months for three years. By age six, Ipek exhibited only mild symptoms of A-T, with no reported adverse effects from the treatment [92].

To date, Milasen and Atipeksen are the only N-of-1 ASOs that have received official or academic publication [92]. However, reports from patient-run sites and news articles suggest additional N-of-1 ASOs are being developed to address unique mutations [92]. Since the promising preliminary findings from these N-of-1 trials, the FDA has introduced specific guidelines for N-of-1 ASO development to streamline the process for patients without existing treatment options [20]. These guidelines include requirements for a non-clinical report involving the results of short-term animal studies; a chemistry, manufacturing, control report outlining the drug’s known structure and properties; as well as a clinical considerations report [92]. By addressing the regulatory challenges inherent in personalized drug development, the FDA’s efforts are expected to pave the way for more N-of-1 ASOs in the future.

Despite the promising advancements in N-of-1 therapies and improvements to regulatory systems for N-of-1 ASOs, financial and ethical challenges remain. A major ethical concern is the inequity of patient access [96,97,98]. Participation in N-of-1 trials is often limited to motivated, well-informed patients who have established connections with experts in the field [99]. These inequities are exacerbated by the financial burden of private-pay requirements and the logistical challenges of travel [100]. Consequently, access to these trials is largely restricted to patients with substantial time and resources. Beyond these access disparities, there are ethical concerns regarding the limited evidence generated through rapid preclinical testing and assumptions about safe dosing [101]. Questions also arise about whether patients can truly provide informed consent, given the uncertainties surrounding the risks of these personalized treatments [99]. As efforts to improve and expand N-of-1 trials progress, it is crucial to address these ethical considerations to promote equitable access and ensure patients are fully informed when making decisions about their participation.

### 2.5. Challenges Associated with ASO Design

Despite the recent successful transitions of many ASOs from the bench to the bedside, there remains considerable room for improvement in their clinical efficacy. For instance, Eteplirsen’s effectiveness in treating DMD remains a subject of ongoing debate [19]. Patients receiving weekly 30 mg/kg doses of Eteplirsen demonstrated functional benefits, such as improved performance on a 6 min walk test and a reduced incidence of ambulation loss [102]. However, the small sample size and reliance on historical control data significantly limit the reliability of these findings. Additionally, treated patients achieved an average of up to only 1% functional dystrophin protein restoration in muscle tissues and showed no improvement in dystrophin expression in cardiac tissues after at least 48 weeks of therapy [19,103]. Whether Eteplirsen can prevent long-term disease progression at this dose remains uncertain.

It has been well established that various factors in ASO design profoundly impact their safety and ability to achieve the intended RNA-targeting effects. These factors include RNA secondary structure, ASO binding energy, chemical modifications, conjugation to alternative molecules, and the specific tissues being targeted [104,105,106,107]. For example, the length of an ASO is a critical factor influencing its efficacy. While increasing the number of nucleotides it is composed of can enhance binding energy and improve overall efficacy at the target site, it may also reduce target specificity [108,109]. Consequently, current guidelines for ASO design generally recommend lengths ranging from 18 to 22 nucleotides to balance efficacy and specificity [108]. Designing effective ASOs is inherently challenging due to the vast number of potential RNA sequences that could theoretically serve the same purpose [21]. This complexity is further compounded by the integration of diverse chemical modifications, which introduce additional hurdles in the design process. For instance, the chirality of the phosphorous atom in the PS backbone has been shown to influence both RNAse H activity and immune stimulation [110,111]. Controlling the chirality of PS ASOs is critical, as it can significantly impact the overall therapeutic efficacy [112,113]. Optimizing ASO design is further complicated when considering the incorporation of different combinations of chemical modifications, which bring additional challenges to the design process [21].

Historically, ASO design has primarily relied on basic methods and guidelines derived from analyses of previously published ASO data, focusing on the relationship between sequence, length, and efficacy [108,114,115,116,117,118,119]. These guidelines typically recommend designing ASOs with a purely reverse complement sequence to the target mRNA and provide recommendations for specific parameters such as ASO length and GC content. However, even when these guidelines are followed, a significant number of potentially effective ASOs can be generated for a single RNA target, necessitating substantial experimental effort to identify the most effective candidates.

Improving ASO design remains a critical barrier to achieving more clinically effective therapies. Studies have demonstrated that optimized ASO sequences can be significantly more effective than current FDA-approved options. For instance, in silico predictive screening tools have been utilized to refine ASO sequences targeting exon 51 of *DMD* pre-mRNA for exclusion [27]. These tools, using statistical modeling approaches, successfully identified a sequence that increased exon 51 skipping by 12-fold compared to Eteplirsen analogs in vitro. However, while numerous in silico tools have been developed to improve ASO sequence design [120,121,122], they currently lack the ability to account for the impact of chemical modifications and other critical factors on ASO efficacy [28].

## 3. Machine Learning-Based Platforms to Improve Antisense Oligonucleotide Design

To address the complexities and challenges of designing ASOs, machine learning-based platforms have been developed to optimize ASO design by leveraging insights from previous studies. For this review, the search terms “Machine Learning” and “Deep Learning”, in combination with “Antisense Oligonucleotide Design”, were used to identify relevant literature in PubMed. The search specifically aimed to identify platforms with the potential to be broadly applicable across diverse mRNA targets, leading to the identification of two recently developed tools: ASOptimizer and eSkipFinder. ASOptimizer was developed for designing ASOs utilizing RNase-H dependent RNA degradation, while eSkip-Finder focuses on optimizing the design of splice-switching ASOs [21,28]. As ASOs gain popularity as a therapeutic strategy, extensive testing has been conducted on ASOs with various RNA targets, with diverse lengths, sequences, and chemical modifications, generating a wealth of experimental data [15,48]. These machine learning-based platforms capitalize on this vast and multifaceted dataset, integrating information about key features, such as sequence composition, chemical modifications, and experimental design, into comprehensive databases. By training predictive models on these data, they can either predict the efficacy of an individual ASO or rank multiple ASOs based on their predictions (Figure 2). This application of machine learning holds immense promise for accelerating ASO development while enhancing their precision and effectiveness, ultimately facilitating their transition from experimental stages to clinically viable therapies [29].

### 3.1. eSkip-Finder

eSkip-Finder (https://eskip-finder.org) is a web-based resource that employs machine learning algorithms to predict the efficacy of ASOs specifically designed for exon skipping [28]. Its database was curated through the manual collection of all available data on exon-skipping ASOs from patents and published literature. To develop a comprehensive predictive model, the database includes information on targeted genes, ASO sequences, chemical modifications, experimental protocols, and observed exon-skipping efficacy. A total of 32 distinct features related to ASOs or experimental design were initially assessed for their impact on predictive accuracy. Features with the highest importance, determined through permutation importance methods [123], were incorporated into the final model [28]. Given the unique effects of different chemical modifications on ASO behavior, separate predictive models were developed for PMOs and 2′MOE ASOs.

The eSkip-Finder platform, involving a support vector regressor predictive model, was trained using data from 566 exon-skipping trials involving 298 novel ASOs targeting *DMD* pre-mRNA. The features selected for the final PMO model included ASO concentration, GC content of exon/intron when blocked by the ASO, predicted binding energy, and predicted accessibility scores of the 3′ end of the target [28]. Similarly, for 2′-MOE ASOs, features such as ASO concentration, ASO GC content, distance from splice acceptor site, remaining GC content of exon when blocked by the ASO, cumulative NI score [124], and predicted target accessibility were incorporated into the model. To evaluate the model’s predictive accuracy, 10% of the *DMD* exon-skipping values were reserved for validation. The test sets for PMO and 2′MOE ASOs yielded R^2^ value of 0.6 and 0.7, respectively [28]. To further assess the platform’s generalizability, eSkip-Finder predictions for three unique PMOs targeting exon 72 of *COL7A1* were compared with their experimental outcomes [125]. The strong correlation between the predictions and experimental results highlight the potential of eSkip-Finder for broader applications beyond *DMD* exon skipping [28].

### 3.2. ASOptimizer

Alternatively, ASOptimizer was developed to enhance the efficacy of ASOs designed to treat diseases via RNase H-dependent RNA degradation [21]. Similarly to eSkip-Finder, ASOptimizer relies on an extensive database compiled from patents and scientific literature. This database encompasses 187,090 entries detailing ASOs targeting 67 unique mRNA targets for degradation. Recognizing the critical importance of both sequence design and chemical modifications, the ASOptimizer platform employs a two-stage optimization approach. The first stage, a sequence optimizer, utilizes a linear factor model to evaluate candidate ASOs. This model analyzes key variables influencing sequence potency, such as GC content, length, the secondary structure of target RNA, and potential off-target effects, to identify the most effective sequences. In the second stage, chemical modifications, including changes to nucleotide structure and inter-nucleotide bonds, are optimized using an edge-augmented graph transformer. This deep graph neural network identifies patterns and relationships within previously reported combinations of chemical modifications, enabling informed decisions about the optimal modifications for ASOs [126].

To determine its effectiveness in improving ASO design, ASOptimizer was applied to optimize ASOs targeting indoleamine 2,3-dioxygenase 1 (*IDO1*) mRNA. The *IDO1* gene encodes an enzyme involved in the kynurenine pathway, which is often upregulated in cancers, enabling the tumor to evade immune detection [127]. Initially, ASOptimizer’s predictive performance was compared to *Sfold*, a widely used in silico ASO design tool employing statistical sampling [120]. Both tools were trained using a subset of 155 ASOs targeting *IDO1* mRNA and tested on the remaining values. ASOptimizer demonstrated superior predictive accuracy, achieving a Pearson correlation of 0.66, compared to *Sfold*’s 0.5 [21]. Next, ASOptimizer was used to identify six optimized 19-nucleotide sequence candidates for improved *IDO1* inhibition. Through in vitro testing, all six ASO sequences, developed with only PS backbone modifications, were successful in reducing *IDO1* expression. Subsequently, the chemical modifications for these six ASOs were optimized in the second stage of the ASOptimizer. Current literature on ASOs targeting *IDO1* predominantly employs LNA modifications with a PS backbone. As such, ASOptimizer introduced diverse combinations of LNA modifications at the ends of the ASO to form LNA-PS gapmers. Interestingly, several candidates lacked the typical gapmer structure, where flanked outer regions are fully modified, and instead featured a combination of modified and unmodified nucleotides at the end. When compared to their unoptimized PS-ASO counterparts, the optimized LNA-PS gapmers demonstrated improved efficacy and reduced cytotoxicity as measured by lactate dehydrogenase levels [21]. These findings validated the effectiveness of ASOptimizer’s chemical engineering stage in enhancing both potency and cellular safety profile of ASOs.

### 3.3. Limitations and Future Directions of Current Machine Learning-Based Platforms

Despite the promising potential of these machine learning-based platforms, they are not without limitations. Both eSkip-Finder and ASOptimizer have been validated using data from only one or two ASO targets, raising questions about the generalizability to alternative RNA targets [28,29]. Although the Pearson correlation and R^2^ values achieved in their test sets indicate their utility in optimizing ASO design [128], there remains considerable room for improvement. Updating and expanding the databases used to train these models could improve their predictive accuracy and robustness [28]. Additionally, more extensive validation of these platforms’ generalizability to alternative targets is essential to bolster confidence in their practical utility. This validation should encompass assessing the platforms’ ability to optimize ASO design for wide range of targets and evaluating their predictive performance for ASOs incorporating alternative chemistries not included in the initial testing. Such efforts would greatly enhance confidence in the reliability and applicability of these platforms for broader use in ASO therapeutics.

Despite promising in vitro data on efficacy and cytotoxicity, these platforms currently do not account for the effectiveness or safety of the ASOs they optimize in vivo. Consequently, ASOs generated through these platforms still require rigorous evaluation in vivo to ensure they achieve their intended effects without adverse outcomes [29]. While ASOs designed using standard guidelines are generally safe at low doses, those optimized by these platforms may incorporate novel combinations of chemical modifications, such as the novel gapmer structures developed through ASOptimizer for *IDO1* inhibition [21]. To further solidify confidence in the clinical utility and safety of ASOs optimized by these platforms, additional in vivo validation studies assessing both efficacy and safety are imperative.

Significant gaps persist in the current landscape of machine learning platforms for ASO design. For instance, no platform has been developed specifically for optimizing ASOs intended for exon inclusion, likely because this mechanism has narrower applicability compared to exon skipping and RNA degradation [129]. However, the increased complexity of exon-inclusion design highlights the potential value of such a platform to enhance sequence optimization [61].

Furthermore, while ASOptimizer effectively integrates a sequence engineering model, it does not address critical factors such as ASO delivery to target tissues in vivo [29]. The suboptimal delivery and cellular uptake of ASOs remain significant barriers to achieving optimal therapeutic efficacy. Notably, ASOs exhibit varying uptake rates across different tissues, and selective tissue targeting has proven challenging when ASOs are delivered independently [39,130,131]. Additionally, ASOs face significant difficulty crossing the blood–brain barrier, necessitating invasive intrathecal injections for targeting tissues within the central nervous system [132]. To address these challenges, various non-viral delivery systems, including lipid nanoparticles and nano-carrier-based approaches, as well as covalent conjugation strategies using molecules such as cell-penetrating peptides, have been explored to enhance tissue-specific delivery and cellular uptake [133,134,135]. As several delivery systems and conjugation with alternative molecules have resulted in promising pre-clinical results [133,134,136], these factors will become an important feature to incorporate into current and future ASO design platforms to improve their therapeutic potential.

Nevertheless, these platforms hold immense promise for advancing ASO design and development. Notably, eSkip-Finder has undergone enhancements to its predicative algorithm since its release. A three-way voting system combining random forest, gradient boosting, and XGBoost was shown to significantly reduce computing costs compared to the original support vector regressor model, while also improving R^2^ values in the test sets analyzed [137]. By further expanding the underlying databases and refining their predictive algorithms, these platforms could revolutionize ASO development. They enable the selection of highly effective ASOs through computational modeling, reducing the need for costly and time-consuming in vitro and in vivo screening experiments. These advancements are anticipated to substantially improve the clinical efficacy of ASOs, particularly in time-sensitive cases such as N-of-1 trials, where rapid development is crucial [92,93]. The ability of these platforms to accelerate the design of therapeutically effective ASOs has the potential to modify disease progression more efficiently and enhance patient outcomes.

## 4. Conclusions

Machine learning holds immense promise in addressing the complexities of designing effective ASOs for rare diseases [29]. Existing platforms have demonstrated success in optimizing ASO sequences and chemical modifications by accounting for a wide range of critical factors [21,28]. However, significant limitations remain, including the need for further validation of their generalizability and addressing gaps in ASO design features that are not yet incorporated into these models [29]. Advancing these machine learning-based platforms to predict the most effective sequences and chemical structures will not only reduce the time and costs associated with early preclinical trials, but also enhance our ability to deliver highly efficient therapies to patients more rapidly.

## Figures and Tables

**Figure 1 genes-16-00185-f001:**
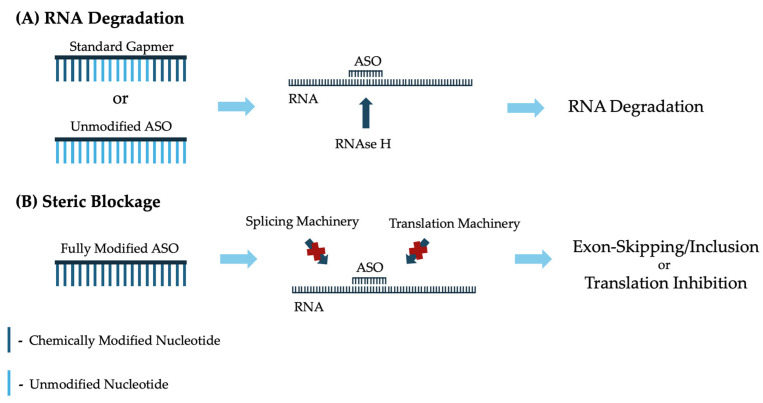
Mechanisms by which ASOs can be utilized to treat genetic diseases and their standard structural configurations to achieve the desired mechanism. Chemically modified nucleotides are depicted in dark blue while unmodified nucleotides are shown in light blue. (**A**) Standard gapmers, consisting of chemically modified nucleotides flanking a central segment of unmodified nucleotides, as well as fully unmodified ASOs, bind to target RNA, recruiting RNase H to induce RNA degradation. (**B**) Fully chemically modified ASOs can bind to target RNA to block splicing or translation machinery, facilitating exon-skipping or inclusion and translation inhibition, respectively.

**Figure 2 genes-16-00185-f002:**
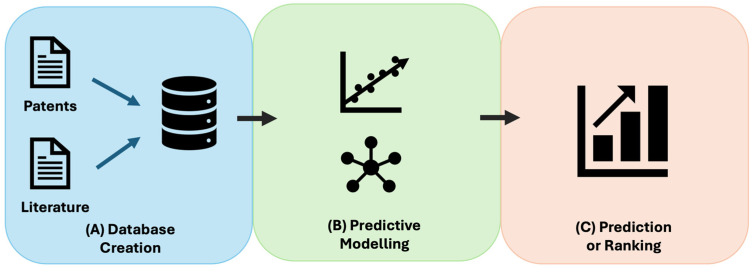
Overview of machine learning-based platforms designed to enhance ASO efficacy through sequence optimization and optimized chemical engineering. (**A**) A comprehensive database is built using data extracted from relevant literature and patents. (**B**) A predictive model is trained on the curated database, leveraging key features associated with ASO performance. (**C**) The trained model predicts the efficacy of individual ASO candidates or ranks multiple ASOs based on their predicted therapeutic potential.

**Table 1 genes-16-00185-t001:** Overview of FDA-approved ASOs, highlighting the diseases they treat, their RNA targets, chemical modifications, and mechanisms of action used to alleviate symptoms.

Therapy Name	Disease	Target RNA	Mechanism of Action	Chemical Modifications
Fomiversen	CMV retinitis	*MIE*	RNA degradation	PS
Mipomersen	HoFH	*APOB*	RNA degradation	PS and 2′-MOE
Eteplirsen	DMD	*DMD* (exon 51)	Splice switching	PMO
Nusinersen	SMA	*SMN2* (exon 7)	Splice switching	PS and 2′-MOE
Inotersen	hATTR	*TTR*	RNA degradation	PS and 2′-MOE
Golodirsen	DMD	*DMD* (exon 53)	Splice switching	PMO
Viltolersen	DMD	*DMD* (exon 53)	Splice switching	PMO
Casimersen	DMD	*DMD* (exon 45)	Splice switching	PMO
Tofersen	ALS	*SOD1*	RNA degradation	PS and 2′-MOE

CMV, cytomegalovirus; HoFH, homozygous familial hypercholesterolemia; DMD, Duchenne muscular dystrophy; SMA, spinal muscular atrophy; hATTR, hereditary transthyretin amyloidosis; ALS, amyotrophic lateral sclerosis.

**Table 2 genes-16-00185-t002:** Overview of ASOs evaluated through N-of-1 clinical trials with published data reporting their results highlighting the diseases they treat, their RNA targets, chemical modifications, and mechanisms of action used to alleviate symptoms.

Therapy Name	Disease	Target	Mechanism of Action	Chemical Modifications
Milasen	Batten Disease	*CLN7* (intron 6)	Splice Switching	PS and 2′-MOE
Atipeksen	HoFH	*ATM* (exon 53)	Splice Switching	PS and 2′-MOE

HoFH, homozygous familial hypercholesterolemia.

## Data Availability

No new data were created or analyzed in this study. Data sharing is not applicable to this article.

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
