# Peer review of "Integrating Machine Learning-Based Approaches into the Design of ASO Therapies"

_genes, 2025, doi:10.3390/genes16020185_

Round 1
Reviewer 1 Report
Comments and Suggestions for Authors
The manuscript "Integrating Machine Learning-Based Approaches into the Design of ASO Therapies" by Lechie and Yokota is a review about the recent advancements of machine learning applied to antisense therapeutical technologies. I must honestly say that this was a very interesting and complete read. Paragraph 2 describes the various application of ASO therapy. Paragraph 3 describes the machine learning platforms to perform ASO design.
I have only a few minor observations.
- The review is not directly discussing one very important parameter and element in ASO design, which is the LENGTH of the designed oligonucleotide. Every approach described has vastly different accuracies whether the oligo is a short RNA (20-30nt, akin to a miRNA) or a longer RNA (akin to a long noncoding RNA). The authors should discuss this important parameter.
- No mention on antisense therapies in the context of anti-pathogen strategies (with the exception of a brief mention on antiretroviral therapies). How can machine learning approaches perform in designing the new generation of RNA-based vaccines and serums?
- While some machine learning approaches are described, there is no discussion of said methods compared to non-ML computational design methods. A reader who is not fully versed with ASO design would be brought directly to a very advanced, niche field of ASO design, without knowing the existence of basic methods (pure reverse complement sequences, 2D structure prediction, et cetera).
- Conclusion pargraph should be number 4, not number 3
Author Response
Response to Reviewer One:
The manuscript "Integrating Machine Learning-Based Approaches into the Design of ASO Therapies" by Lechie and Yokota is a review about the recent advancements of machine learning applied to antisense therapeutical technologies. I must honestly say that this was a very interesting and complete read. Paragraph 2 describes the various application of ASO therapy. Paragraph 3 describes the machine learning platforms to perform ASO design. I have only a few minor observations.
Comment 1: The review is not directly discussing one very important parameter and element in ASO design, which is the LENGTH of the designed oligonucleotide. Every approach described has vastly different accuracies whether the oligo is a short RNA (20-30nt, akin to a miRNA) or a longer RNA (akin to a long noncoding RNA). The authors should discuss this important parameter.
Response: Thank you for commenting on the need for a better discussion of the role of length in the ASO’s overall efficacy. A discussion on the role of ASO length has been included on lines 310-315, that reads: “For example, the length of an ASO is a critical factor influencing its efficacy. While increasing the number of nucleotides it is composed of can enhance binding energy and improve overall efficacy at the target site, it may also reduce target specificity [103,104]. Consequently, current guidelines for ASO design generally recommend lengths ranging from 18 to 22 nucleotides to balance efficacy and specificity [103].”. Longer RNA-based therapeutics are outside of the scope of this paper.
Comment 2: No mention on antisense therapies in the context of anti-pathogen strategies (with the exception of a brief mention on antiretroviral therapies). How can machine learning approaches perform in designing the new generation of RNA-based vaccines and serums?
Response: Thank you for the thoughtful comment. These RNA-based vaccines, and alternative RNA-based therapeutics, are outside of the scope of this paper due to differences in their design and function. To clarify this in the manuscript, the following has been included on lines 71-75, that reads: “Apart from a brief introduction covering the history of the first FDA-approved ASO, the focus is specifically on short ASOs targeting genetic diseases. Alternative RNA-based therapeutics, such as siRNAs and mRNA vaccines, operate on distinct design principles and mechanisms of action and are therefore beyond the scope of this review.”
Comment 3: While some machine learning approaches are described, there is no discussion of said methods compared to non-ML computational design methods. A reader who is not fully versed with ASO design would be brought directly to a very advanced, niche field of ASO design, without knowing the existence of basic methods (pure reverse complement sequences, 2D structure prediction, et cetera).
Response: We appreciate your valuable comment and have included a description of basic methods for ASO design on lines 325-332, that reads: “Historically, ASO design has primarily relied on basic methods and guidelines derived from analyses of previously published ASO data, focusing on the relationship between sequence, length, and efficacy [103,105–110]. These guidelines typically recommend designing ASOs with a purely reverse complement sequence to the target mRNA and provide recommendations for specific parameters such as ASO length and GC content. However, even when these guidelines are followed, a significant number of potentially effective ASOs can be generated for a single RNA target, necessitating substantial experimental effort to identify the most effective candidates.” Additionally, Line 339 has been updated to provide additional description of in silico tools that reads “These tools, using statistical modeling approaches, successfully identified a sequence that increased exon 51 skipping by 12-fold compared to Eteplirsen analogs in vitro.”
Comment 4: Conclusion paragraph should be number 4, not number 3
Response: Thank you for commenting on the incorrect number used for the conclusion paragraph. This has been updated to reflect that the conclusion is section 4.
Reviewer 2 Report
Comments and Suggestions for Authors
In this review article, the authors address an important and emerging area of research, highlighting the potential of machine learning (ML) to optimize antisense oligonucleotide (ASO) therapies, which is crucial for rare diseases. The study has provided an extensive review of ASO design mechanisms, FDA-approved therapies, and emerging ML platforms such as eSkip-Finder and ASOptimizer. The focus of the review on integrating machine learning into ASO design adds significant value by suggesting novel methodologies to address longstanding challenges in ASO therapeutic development.
However, there are some concerns on this study:
1, The platforms (eSkip-Finder and ASOptimizer) are validated using data from only one or two ASO targets. This raises questions about their applicability to a broader range of diseases and targets. The paper could benefit from discussing more diverse case studies or ongoing efforts to generalize the models. The authors could explore a wider range of RNA targets to test and validate their ML models, demonstrating broader applicability and enhancing confidence in their use.
2, The study leans heavily on in vitro data to support the efficacy of ML-designed ASOs. However, translating these results to in vivo or clinical settings remains uncertain. These defects should be addressed by the authors.
3, While the paper extensively reviews ASO sequence optimization, it overlooks the complexities of ASO delivery. The delivery method plays a critical role in therapeutic success, and ignoring this limits the real-world applicability of ML-designed ASOs. The authors should discuss more and address these issues properly. Incorporating ML approaches to optimize ASO delivery mechanisms and tissue targeting could significantly improve therapeutic outcomes.
4, Although the study touches on cytotoxicity in vitro, it does not adequately address potential in vivo safety issues that might arise from ML-designed ASOs. This omission weakens the practical applicability of the platforms.
5, The paper mentions the success of N-of-1 trials but lacks in-depth discussion about regulatory hurdles, patient access, and ethical implications of personalized ASO therapies. Further discussion to address these aspects would provide a more holistic view of the challenges ahead.
Author Response
Response to Reviewer Two:
In this review article, the authors address an important and emerging area of research, highlighting the potential of machine learning (ML) to optimize antisense oligonucleotide (ASO) therapies, which is crucial for rare diseases. The study has provided an extensive review of ASO design mechanisms, FDA-approved therapies, and emerging ML platforms such as eSkip-Finder and ASOptimizer. The focus of the review on integrating machine learning into ASO design adds significant value by suggesting novel methodologies to address longstanding challenges in ASO therapeutic development.
Comment 1: The platforms (eSkip-Finder and ASOptimizer) are validated using data from only one or two ASO targets. This raises questions about their applicability to a broader range of diseases and targets. The paper could benefit from discussing more diverse case studies or ongoing efforts to generalize the models. The authors could explore a wider range of RNA targets to test and validate their ML models, demonstrating broader applicability and enhancing confidence in their use.
Response: We greatly appreciate the thoughtful input. At this time, further testing of the discussed platforms to test additional targets has not been published. To clarify the importance of future efforts to confirm the generalizability of these platforms, the following has been included on lines 445-449: “This validation should encompass assessing the platforms’ ability to optimize ASO design for wide range of targets and evaluating their predictive performance for ASOs incorporating alternative chemistries not included in the initial testing. Such efforts would greatly enhance confidence in the reliability and applicability of these platforms for a broader use in ASO therapeutics.”
Comment 2: The study leans heavily on in vitro data to support the efficacy of ML-designed ASOs. However, translating these results to in vivo or clinical settings remains uncertain. These defects should be addressed by the authors.
Response: Thank you for the valuable suggestion to include a discussion about the current limitations of the platforms due to their lack of vivo efficacy data. The following discussion has been included on lines 450-459 that reads: “Despite promising in vitro data on efficacy and cytotoxicity, these platforms currently do not account for the effectiveness and safety of the ASOs they optimize in vivo. Consequently, ASOs generated through these platforms still require rigorous evaluation in vivo to ensure they achieve their intended effects without adverse outcomes [29]. While ASOs designed using standard guidelines are generally safe at low doses, those optimized by these platforms may incorporate novel combinations of chemical modifications, such as the novel gapmer structures developed through ASOptimizer for IDO1 inhibition [21]. To further solidify confidence in the clinical utility and safety of ASOs optimized by these platforms, additional in vivo validation studies assessing both efficacy and safety are imperative.”
Comment 3: While the paper extensively reviews ASO sequence optimization, it overlooks the complexities of ASO delivery. The delivery method plays a critical role in therapeutic success, and ignoring this limits the real-world applicability of ML-designed ASOs. The authors should discuss more and address these issues properly. Incorporating ML approaches to optimize ASO delivery mechanisms and tissue targeting could significantly improve therapeutic outcomes.
Response: We appreciate your valuable comment to provide a discussion on the importance of delivery mechanisms and tissue targeting in ML platforms. The discussion on ASO delivery has been updated to better address the importance of these factors on lines 466-480, that reads: “Furthermore, while ASOptimizer effectively integrates a sequence engineering model, it does not address critical factors such as ASO delivery to target tissues in vivo [29]. The suboptimal delivery and cellular uptake of ASOs remain significant barriers to achieving optimal therapeutic efficacy. Notably, ASOs exhibit varying uptake rates across different tissues, and selective tissue targeting has proven challenging when ASOs are delivered independently [123–125]. Additionally, ASOs face significant difficulty crossing the blood-brain barrier, necessitating invasive intrathecal injections for targeting tissues within the central nervous system [126]. To address these challenges, various non-viral delivery systems, including lipid nanoparticles and nano-carrier-based approaches, as well as covalent conjugation strategies using molecules such as cell-penetrating peptides (CPPs), have been explored to enhance tissue-specific delivery and cellular uptake [127–129]. As several delivery systems and conjugation with alternative molecules have resulted in promising pre-clinical results [127,128,130], these factors will become an important feature to incorporate into current and future ASO design platforms to improve their therapeutic potential.”
Comment 4: Although the study touches on cytotoxicity in vitro, it does not adequately address potential in vivo safety issues that might arise from ML-designed ASOs. This omission weakens the practical applicability of the platforms.
Response: Thank you for the thoughtful comment to include a discussion on the potential in vivo safety issues that may arise from the ASOs optimized through the discussed platforms. The following discussion on lines 450-459 has been included, that reads: “Despite promising in vitro data on efficacy and cytotoxicity, these platforms currently do not account for the effectiveness and safety of the ASOs they optimize in vivo. Consequently, ASOs generated through these platforms still require rigorous evaluation in vivo to ensure they achieve their intended effects without adverse outcomes [29]. While ASOs designed using standard guidelines are generally safe at low doses, those optimized by these platforms may incorporate novel combinations of chemical modifications, such as the novel gapmer structures developed through ASOptimizer for IDO1 inhibition [21]. To further solidify confidence in the clinical utility and safety of ASOs optimized by these platforms, additional in vivo validation studies assessing both efficacy and safety are imperative.”
Comment 5: The paper mentions the success of N-of-1 trials but lacks in-depth discussion about regulatory hurdles, patient access, and ethical implications of personalized ASO therapies. Further discussion to address these aspects would provide a more holistic view of the challenges ahead.
Response: In response to your valuable suggestion to include an in-depth discussion about the existing barriers and ethical concerns of N-of-1 trials, we have included a discussion on lines 281-294, that reads: “Despite the promising advancements in N-of-1 therapies and improvements to regulatory systems for N-of-1 ASOs, financial and ethical challenges remain. A major ethical concern is the inequity of patient access [96–98]. Participation in N-of-1 trials is often limited to motivated, well-informed patients who have established connections with experts in the field [99]. These inequities are exacerbated by the financial burden of private-pay requirements and the logistical challenges of travel [100]. Consequently, access to these trials is largely restricted to patients with substantial time and resources. Beyond these access disparities, there are ethical concerns regarding the limited evidence generated through rapid preclinical testing and assumptions about safe dosing [101]. Questions also arise about whether patients can truly provide informed consent, given the uncertainties surrounding the risks of these personalized treatments [99]. As efforts to improve and expand N-of-1 trials progress, it is crucial to address these ethical considerations to promote equitable access and ensure patients are fully informed when making decisions about their participation.
Reviewer 3 Report
Comments and Suggestions for Authors
Rare diseases are an important problem not only from the therapeutical point of view but also ethical. The low number of patients and less interest in pharmaceutical companies make them less economically attractive. Fortunately, antisense therapy appeared in the late 70s last century. The pivotal role of this strategy play phosphorothioates. Despite the new oligo derivatives appearing on the market, the PS market is still attractive. PS was first mentioned by Fritz Eckstein. Since that time, ASO got, at the beginning of the XXI century, the FDA approval. The second important point for new therapy of orphan diseases are the full genome sequence discovery. Bot of the below leads to the possibility of fast target sequence assignment. However, an effective tool for this challenge is necessary. Therefore, the article entitled Integrating Machine Learning-Based Approaches into the Design of ASO Therapies can be valuable for the broad scientific community due to its scientific landscape in the fields.
I have some suggestions: the graphical representation of the ASO building units should be the focus. Moreover, the chirality of phosphorothioates must be discussed, as well as the Stec group publications about PS.
From the editorial point of view, the article is well-written and readable – it is the one I read with pleasure. The references have been almost well-selected and cited.
In conclusion, I can recommend this article for publication after answering my questions.
Author Response
Response to Reviewer Three:
Rare diseases are an important problem not only from the therapeutical point of view but also ethical. The low number of patients and less interest in pharmaceutical companies make them less economically attractive. Fortunately, antisense therapy appeared in the late 70s last century. The pivotal role of this strategy play phosphorothioates. Despite the new oligo derivatives appearing on the market, the PS market is still attractive. PS was first mentioned by Fritz Eckstein. Since that time, ASO got, at the beginning of the XXI century, the FDA approval. The second important point for new therapy of orphan diseases are the full genome sequence discovery. Bot of the below leads to the possibility of fast target sequence assignment. However, an effective tool for this challenge is necessary. Therefore, the article entitled Integrating Machine Learning-Based Approaches into the Design of ASO Therapies can be valuable for the broad scientific community due to its scientific landscape in the fields. From the editorial point of view, the article is well-written and readable – it is the one I read with pleasure. The references have been almost well-selected and cited. In conclusion, I can recommend this article for publication after answering my questions.
Comment 1: the graphical representation of the ASO building units should be the focus.
Response: Thank you for your valuable suggestion to focus on the ASO building units in the graphical representation. The caption for Figure 1 has been updated to provide a greater focus on the ASO building units and now reads: “Figure 1. Mechanisms by which ASOs can be utilized to treat genetic diseases and their standard structural configurations to acheive the desired mechanism. Chemically modified nucleotides are depicted in dark blue while unmodified nucleotides are shown in light blue. A) standard gapmers, consisting of chemically modified nucleotides flanking a central segment of unmodified nucleotides, as well as fully unmodified ASOs, bind to target RNA, recruiting RNase H to induce RNA degradation. B) Fully chemically modified ASOs can bind to target RNA to block splicing or translation machinery, facilitating exon-skipping or inclusion and translation inhibition, respectively.”
Comment 2: Moreover, the chirality of phosphorothioates must be discussed, as well as the Stec group publications about PS
Response: We appreciate your thoughtful request for the inclusion of a discussion on the role of chirality of phosphorothioates in the manuscript. The following discussion has been included on lines 319-322, that reads: “. For instance, the chirality of the phosphorous atom in the PS backbone has been shown to influence both RNAse H activity and immune stimulation [110,111]. Controlling the chirality of PS ASOs is critical, as it can significantly impact the overall therapeutic efficacy [112,113].”
Reviewer 4 Report
Comments and Suggestions for Authors
An interesting article has been sent for my review.
The article was generally prepared correctly; I only have a few minor comments.
I have a question: although it is not a systematic review, what keywords did the authors use to review the databases? What databases did they review, the criteria for selecting articles, and what were the exclusion criteria? What time frames of the conducted studies did they take into account? Including a few sentences on this topic in the publication would be helpful.
I suggest an explanation of the abbreviations be inserted under Table 1 - primarily the explanation of diseases.
I found several similar articles, such as:
Lin S, Hong L, Wei DQ, Xiong Y. Deep learning facilitates efficient optimization of antisense oligonucleotide drugs. Mol Ther Nucleic Acids. 2024 May 16;35(2):102208. doi: 10.1016/j.omtn.2024.102208. PMID: 38803420; PMCID: PMC11129084.
Zhu A, Chiba S, Shimizu Y, Kunitake K, Okuno Y, Aoki Y, Yokota T. Ensemble-Learning and Feature Selection Techniques for Enhanced Antisense Oligonucleotide Efficacy Prediction in Exon Skipping. Pharmaceutics. 2023 Jun 24;15(7):1808. doi: 10.3390/pharmaceutics15071808. PMID: 37513994; PMCID: PMC10384346.
Hence, I suggest that the authors refer to existing review articles on this topic and emphasize the novelty of their article.
Author Response
Response to Reviewer Four:
An interesting article has been sent for my review.
The article was generally prepared correctly; I only have a few minor comments.
Comment 1: I have a question: although it is not a systematic review, what keywords did the authors use to review the databases? What databases did they review, the criteria for selecting articles, and what were the exclusion criteria? What time frames of the conducted studies did they take into account? Including a few sentences on this topic in the publication would be helpful.
Response: We appreciate the valuable comment to include a few sentences on the keywords and criteria for reviewing the databases. We have included a discussion on lines 347-351, that reads: “. For this review, the search terms “Machine Learning” and “Deep Learning,” in combination with “Antisense Oligonucleotide Design” were used to identify relevant literature in PubMed. The search specifically aimed to identify platforms with the potential to be broadly applicable across diverse mRNA targets, leading to the identification of two recently developed tools: ASOptimizer and eSkipFinder.”
Comment 2: I suggest an explanation of the abbreviations be inserted under Table 1 - primarily the explanation of diseases.
Response: We appreciate the valuable suggestion to include the abbreviations of the listed diseases under Table 1. A table footer for Table 1 and Table 2 have been included to clarify the abbreviations used for the diseases.
Comment 3: I found several similar articles, such as:
Lin S, Hong L, Wei DQ, Xiong Y. Deep learning facilitates efficient optimization of antisense oligonucleotide drugs. Mol Ther Nucleic Acids. 2024 May 16;35(2):102208. doi: 10.1016/j.omtn.2024.102208. PMID: 38803420; PMCID: PMC11129084.
Zhu A, Chiba S, Shimizu Y, Kunitake K, Okuno Y, Aoki Y, Yokota T. Ensemble-Learning and Feature Selection Techniques for Enhanced Antisense Oligonucleotide Efficacy Prediction in Exon Skipping. Pharmaceutics. 2023 Jun 24;15(7):1808. doi: 10.3390/pharmaceutics15071808. PMID: 37513994; PMCID: PMC10384346.
Hence, I suggest that the authors refer to existing review articles on this topic and emphasize the novelty of their article.
Response: Thank you for your thoughtful comment. The Zhu et al., 2023 study is an original research article that is discussed on lines 487-491, that reads “Notably, eSkip-Finder has undergone enhancements to its predicative algorithm since its release. A three-way voting system combining random forest, gradient boosting, and XGBoost was shown to significantly reduce computing costs compared to the original support vector regressor model, while also improving R2 values in test sets analyzed [140].” To reference the Lin et al. (2024) review and clarify the novelty of this review article, the following has been included on lines 67-71, that reads: “A previous review primarily focused on providing an overview of a single machine learning-based platform [29]. In contrast, this review provides a comprehensive overview of ASOs, their application in N-of-1 trials, the challenges associated with optimizing ASO design, and current machine learning-based platforms that have been developed in an attempt to improve ASO design”